# The Study of Enteromorpha-Based Reinforcing-Type Flame Retardant on Flame Retardancy and Smoke Suppression of EPDM

**DOI:** 10.3390/polym15010055

**Published:** 2022-12-23

**Authors:** Peipei Sun, Ziwen Zhou, Licong Jiang, Shuai Zhao, Lin Li

**Affiliations:** 1Advanced Materials Institute, State Key Laboratory of Biobased Material and Green Papermaking, Qilu University of Technology (Shandong Academy of Sciences), Jinan 250014, China; 2Key Lab of Rubber-Plastics, Ministry of Education/Shandong Provincial Key Lab of Rubber-Plastics, School of Polymer Science and Engineering, Qingdao University of Science and Technology, Qingdao 266042, China

**Keywords:** Enteromorpha, reinforcing, flame retardant

## Abstract

Enteromorpha, as a waste from marine pollution, brings great pressure to environmental governance every year, especially for China. Under the premise of a shortage of industrial materials, taking appropriate measures can turn waste into wealth, which will benefit us a lot. In this work, a bio-based reinforcing-type flame retardant based on Enteromorpha is designed. The designed Enteromorpha-based flame retardant system (AEG) mainly focuses on the reinforcing and flame retardant effects on ethylene-propylene-diene tripolymer (EPDM). For the AEG system, ammonium polyphosphate (APP) serves as both the acid source and the gas source; the simple hybrid material (GN) produced by loading graphene (GE) and Enteromorpha (EN) using tannic acid (TA) as a regulator serves as an acid source and a carbonizing source. The results show that when 40 phr AEG is added, the LOI of EPDM/AEG40 reaches 32.5% and the UL-94 reaches the V-0 level. The PHRR and THR values of EPDM/AEG40 are 325.9 kW/m^2^ and 117.6 MJ/m^2^, respectively, with decrements of 67.3% and 29.7%, respectively, compared with the results of neat EPDM composite. Especially, the TSP and TSR values of EPDM/AEG40 are reduced from 15.2 m^2^ of neat EPDM to 9.9 m^2^ with a decrement of 34.9% and reduced from 1715.2 m^2^/m^2^ of neat EPDM to 1124.5 m^2^/m^2^ with a decrement of 34.4%, indicating that AEG is effective in flame retardancy and smoke suppression. Meanwhile, the tensile strength and tear strength of EPDM/AEG composites are much higher than neat EPDM, therefore, with the future development of innovate reinforcing-type flame-retardant Enteromorpha, the application of Enteromorpha in the polymer flame-retardant field will surely usher in bright development.

## 1. Introduction

In recent years, with the prices of raw materials rising, the search for alternative materials from nature has become increasingly serious, especially for the rubber industry. Rubber, as an important commercial material, is inseparable from our lives. For engineering applications, adding fillers to rubber is necessary to improve its modulus, strength, wear resistance and fatigue resistance and other properties. Since 2008, due to the high nitrogen levels caused by fertilizers, the suddenly immense outbreak of Enteromorpha as a kind of green algae in the east coast of China caused severe environmental problems and has threatened coastal aquaculture [1,2,3]. Qingdao is a beautiful city in China where every year a large amount of Enteromorpha need to be cleaned off the coast to maintain normal shipping, safe fishing and clean swimming. In 2015, more than 70,000 tons of seaweed were treated [4]. However, everything has two sides. Enteromorpha mainly contains cellulose and polysaccharides (43.4–60.2%), crude protein (9.0–14.0%), ether extract (2.0–3.6%), ash (32.0–36.0%), as well as some fatty acid [5,6]. As an important phytochemical of green algae, Enteromorpha prolifera polysaccharides have been extensively studied in explosive growth and have been proved to have various physiological and biological activities, including anti-oxidant [7], immunomodulation [8], anti-bacteria [9], anti-hyperlipidemia [10], anti-tumor [11], anti-cancer [12], anti-viral [13], and anti-coagulant [14] properties; it also regulates gut microbiota [15]. As a researcher in the field of composite materials, Enteromorpha's thin, silk-like microstructure and malleable carbonized performance have been of extreme interest to me [16]. In the rubber industry, fiber with a certain aspect ratio is usually used to improve the dimensional stability of unvulcanized rubber and the comprehensive properties of vulcanized rubber by increasing the green strength of the rubber matrix [17]. Fiber with a proper orientation can change the tensile strength of rubber considerably so that, at low elongation, it is impossible to achieve a sharp rise in stress with particle filler [18,19,20]. This feature is very useful if large deformation is undesirable [17,21]. Enteromorpha, with its thin, silk-like microstructure, may have the potential to replace traditional rubber-reinforcing fillers, alleviating the serious problem of material shortages. Nevertheless, the malleable carbonized performance of Enteromorpha with typical flame-retardant elements (phosphorus and nitrogen) gives it a certain application potential in the field of flame retardants. As is known to all, the mechanism of flame retardancy can be divided into gas phase and condensed phase flame retardants [22]. More precisely, the main mechanism of condensed phase flame retardants is promotion of the formation of char residues that protect the underlying matrix from further combustion [6]. In our previous work, an intumescent flame-retardant system was successfully designed by applying tannic-acid-functionalized graphene combined with ammonium polyphosphate. The synergistic effects of the designed intumescent flame retardant on the flame retardancy and smoke suppression performances of natural rubber resulted in very satisfactory performances.

In this work, we design another efficient bio-based reinforcement-type intumescent flame-retardant system (AEG) based on Enteromorpha, which is environmentally friendly and has potential for introduction into the polymer flame retardant field. In the AEG system, ammonium polyphosphate (APP) is used as both an acid source and a gas source, and the hybrid material (GN), prepared by GE and EN using TA as a regulator, is the acid source and the carbon source. AEG flame retardant system is mainly focused on the enhancement and flame retardancy effects on ethylene-propylene-diene rubber (EPDM). The microstructure and thermal degradation of EN and AEG before and after combustion are analyzed. The thermal degradation and combustion properties of EPDM/AEG composites were investigated by thermogravimetric and microscale combustion calorimetric analysis, Limited Oxygen Index and UL-94 vertical burning test. The microscopic quality of intumescent char is examined using SEM-EDS, FTIR and Raman. Based on the analysis of thermal decomposition behavior of polymer composites and the characteristics of carbon, the flame-retardant mechanism was put forward.

## 2. Experimental and Materials 

### 2.1. Materials

Ammonium polyphosphate (APP, type is TF-201 with 1500 polymerization degree) was generously supplied by Taifeng New Flame Retardant Co., Ltd., Deyang, China. Graphene (GE) was supplied by the Sixth Element (Changzhou, China) Material Technology Co., Ltd. (Changzhou, China). Enteromorpha was sourced from the southeast coast of Shandong Peninsula in 2017. Tannic acid (TA, analytical grade) was purchased from Aladdin Technology Co., Ltd. (Shanghai, China). EPDM (8550C, the third monomer content is 5%) was purchased from Alangxingke High Performance Elastomer Co., Ltd. (Shanghai, China). Other reagents, including sulfur, 2-mercaptobenzothiazole (M), Tetramethylthiuram disulfide (TT), zinc oxide (ZnO) and Stearic acid (SA), were all industrial grade and kindly supplied by Qingdao Topsen Chemical Co., Ltd. (Qingdao, China). 

### 2.2. Characterization

Raman spectra analyses were recorded using a Bruker FRS-100S with high-resolution and a CCD detector in the wavelength range of 600 to 2500 cm^−1^. A Bruker Vertex 70 Fourier Transform Infrared Spectrometer was used to characterize the infrared spectra of modified fillers. JSM-6700F SEM-EDX (Electronics Corp. Tokyo, Japan) was used to characterize the morphology of the residue char, for which the samples should be coated with gold for a certain time. Thermal performances were carried out using a TGA-7 instrument (Perkine Elmer Company, Waltham, MA, USA), with a room temperature to 800 °C at 10 °C/min heating rate in N_2_ atmosphere. Dynamic mechanical performances were obtained on a NETZSCH DMA242 machine with tensile mode and a temperature increment of 3 °C·min^−1^ at a fixed frequency of 1 Hz and a fixed strain of 0.5%. Limited Oxygen Index was performed on a HC-2 oxygen index meter (Analysis Instrument Company, Jiangning, China) making a reference to ASTM D2863. Vertical Burning Test, a typical combustion test, was performed on a CFZ-1 vertical burning instrument (Analysis Instrument Co., Jiangning, China) according to ASTM D3801-2010. The heat and smoke index were analyzed using the Cone Calorimeter (Fire Testing Technology, West Sussex, UK) exposed to 35 kW/m^2^ external heat flux referring to ASTM E1354/ISO 5660. Mechanical property tests, including tensile and tear, were carried out using an AI-7000S Universal Material Tester, for which a dumbbell specimen under 500 mm·min^−1^ tensile speed was employed according to ISO 528:2009. The optimum curing temperature of the EPDM composites was 155 °C determined by an ALPHA MDR2000 UCAN rheometer.

### 2.3. Preparation of GN

A wall breaker was used to superfine EN for 30 min at room temperature. GE and EN with varied mass ratios were added into a vial containing 100 mL TA aqueous solution with a fixed concentration of TA, shown in Appendix A. The mixture was then mixed at high speed for 30 min to prepare GN. The mentioned reaction product was thoroughly filtered and washed with deionized water in order to completely remove unreacted TA. The resulting filter cake was dried at 70 °C for 72 h in the oven. Thus, GN powder was obtained.

### 2.4. Preparation of Composite Materials

Flame-retardant AEG was prepared by compounding the prepared GN with APP in a certain proportion (Appendix A), and EPDM composites were charged in a 200 mL Banbury mixer at 60 rpm rotor speed under 120 °C; the formula is shown in Table 1. The EPDM was firstly fed into the mixer for premixing for 2 min. SA (1 phr), zinc oxide (5 phr) and N330 (40 phr) were then added in sequence, and the compounding ingredients were mixed for another 8 min. Then, the compounds were discharged onto a two-roll mill, and sulfur (1.5 phr), M (0.5 phr) and TT (1.5 phr) at were added 40 °C and mixed for another 5 min to achieve a sheet with a 2 mm thickness. The compound sheets were cut and discharged onto a compression molding machine at 165 °C for a time equal to the optimum cure time. The sheet was then cooled at room temperature and named as EPDM/AEG, as shown in Figure 1. The sheets were pressed in a compression molding machine at 165 °C for a time equal to the optimal curing time and under a pressure of 3.94 × 10^4^ kg/m^2^. For comparison, the reference flame retardants are a combination of APP and EN (AE) and a combination of APP and GE (AG); the reference composites are neat EPDM, EPDM/AE and EPDM/AG, which were prepared using the same procedure as EPDM/AEG.

## 3. Results and Discussion

### 3.1. The Particle Size Distribution and Zeta Potential of EN

Ultra-fineness is a common flame-retardant modification method that reduces the particle size of the flame retardant to improve the dispersity in the polymer matrix through physical grinding or chemical modification. Through comparing the influence of the mechanical grinding pretreatment method on the distribution of the EN particle size, it can be seen from Figure 1a that the average particle size of EN after the manual grinding treatment is 5218 nm, and that after the mechanical grinding treatment is 2245 nm. Compared with manually ground EN, mechanically ground EN has a narrower particle size distribution and a higher fineness, which gives the EN fibers a better dispersity in the rubber matrix. Moreover, the Zeta potential mainly characterizes the interaction between colloids and electrolytes. The higher the absolute value of the Zeta potential, the more stable the dispersion of the system. The lower the Zeta potential value, the more easily the filler aggregates and condenses. From Figure 1b, it can be seen that the mechanically ground EN has a higher Zeta potential, indicating that it significantly improves the bonding infiltration interaction at the filler–matrix interface.

### 3.2. Thermal Stability of EN and AEG40

TGA is used to evaluate the thermal stability of materials, as shown in Figure 2, Appendix A and Table 2. EN has two obvious peaks at 92.3 °C and 242.5 °C. The appearance of T_1max_ is the decomposition of water and small molecules in the EN biomass. T_2max_ corresponds to the pyrolysis of oxygen-containing functional groups in EN [1,6]. The residual carbon content of EN is relatively high, which can reach 38.1%. Due to the addition of APP, the T_-5%_ of AEG40 has increased significantly. Because APP decomposes and releases incombustible gas, water and ammonia, the T_1max_ and T_3max_ of AEG40 correspond to the decomposition of polyphosphoric acid to metaphosphoric acid and pyrophosphoric acid to finally generate phosphorous ash. The T_1max_ of AEG40 is higher than the processing temperature of general rubber (150 °C~160 °C), indicating that the flame retardant can maintain good thermal stability during rubber processing without decomposition, and when the rubber burns at 350 °C~450 °C, the flame retardant can quickly decompose to play a flame-retardant effect.

Hongsheng Liu et al. [23] have studied that Enteromorpha is not suitable as a filler for conventional polymers, especially for rubber composites, since it is not only is hydrophilic, but also contains some moisture. This moisture is not easy to remove completely, which is due to Enteromorpha growing in salt water. The presence of water is fatal to the vulcanization of rubber, and the vulcanization results of EPDM composites (Table 3) show that Enteromorpha has no effect on the vulcanization and scorching time of EPDM because the important parameters t_c10_ and t_c90_ have not fundamentally changed. The results show that Enteromorpha is almost completely dried.

### 3.3. Flame-Retardant Properties of EPDM Rubber Composites

Neat EPDM burns violently after being ignited, accompanied by severe melting drips. As shown in Table 4, the LOI data show that neat EPDM is flammable because its LOI value is 24.1%. The digital photo of the carbon residue of neat EPDM (Figure 3) directly shows the sparse carbon residue with less carbonization. With the further addition of flame retardants, the LOI value and UL-94 rating of composite materials are elevated. When 40 phr AEG is added, the LOI of EPDM/AEG40 reaches 32.5% and the UL-94 reaches V-0 level. It is worth noting that EPDM/AG40 has the phenomenon of melt dripping. The LOI is 29.5% and the UL-94 level is V-2. EPDM/AE40 has no melt dripping due to the addition of EN, its LOI reaches 31.2% and its UL-94 level reaches V-1. This indicates that EN is of great help in solving melting dripping. We hypothesize that the high carbon-forming property of EN contributes to the formation of a tight protective carbon layer. Nevertheless, the synergistic effect of EN and GE is a good solution to upgrade the flame retardancy grade of EPDM.

To test our conjecture, the cone calorimeter test and a digital photo of the carbon residue of EPDM composites are further studied, as shown in Figure 3. Figure 3 shows that the char residue becomes much denser with the increase in the added AEG. Compared with EPDM/AG40, the amount of carbonization on EPDM/AE40 is much higher, and the char residue is much denser and more complete. This is more pronounced when EN and GE effectively hybridize, as the AEG system can effectively promote the formation of strong carbonization through increasing the degree of aromatic crosslinking. As the main char-forming agent, EN plays an important role in improving the flame-retardant performance. These results can also be verified by thermal analysis. 

To further study the combustion properties of the AEG system, a more scientific cone calorimeter test was used to evaluate the combustion performance. The burning time after combustion (TTI), the peak heat release rate (PHRR) and total heat release rate (THR), the smoke generation rate (SPR), the total smoke rate (TSR), the total smoke generation (TSP) and the mass residue parameters are shown in Figure 4; the corresponding data are shown in Table 5. Heat is an indispensable stimulant and maintenance factor in a combustion process. For EPDM/AEG composites, AEG flame-retardant systems improve the flame-retardant performances of EPDM mainly by reducing the amount and rate of heat released from combustion. As shown in Figure 4a,b, the peak HRR and THR values have been proven to be key parameters for characterizing the fire safety of polymers. With the increase in the amount of AEG, EPDM/AEG composites show a continuous decrease in HRR and THR. The PHRR and THR values of EPDM/AEG40 are 325.9 kW/m^2^ and 117.6 MJ/m^2^, respectively, showing a decrement of 67.3% and 29.7%, respectively, compared with the results of the neat EPDM composite. The AEG systems also show superior reduction in combustion heat compared to traditional APP flame retardants. For example, the PHRR and THR values of EPDM/AEG40 reduce by 14.3% and 2.4%, respectively, compared with EPDM/APP40 with 40phr APP. In addition, based on 40 phr APP, AE and GE differ greatly in the degree to which they reduce the amount of generated heat as well as the heat generation rate. For EPDM/AE40, the amount of heat generated and the heat generation rate are 300.75 kW/m^2^ of PHRR and 122.1 MJ/m^2^ of THR, which are21.6% and 15.7% lower than EPDM/AG40 (383.4 kW/m^2^ of PHRR and 144.8 MJ/m^2^ of THR, respectively). These results show that EG contributes significantly to the flame retardancy of GN hybrid materials, especially in reducing combustion heat, which is ascribed to the formation of strong, dense and protective char residue layers.

Usually, the fire performance index (FPI) [24] and the fire growth index (FIGRA) [25] are used for judging the fire hazard of the studied AEG system more clearly. FPI is defined as the proportion of TTI to PHRR, and there is a certain correlation between the FPI value of a material and its time to flashover. A reduced FPI value means a shorter time to flashover. FIGRA is defined as the proportion of PHRR to the peak HRR time. A larger FIGRA value means a shorter time to reach the peak HRR time and the greater the fire hazard is for the materials. The FPI and FIGRA values of the studied composites have been shown in Table 5. Based on 40 phr of the main flame retardant APP, the synergistic flame retardancy and smoke suppression effects of GN, EN and GE on EPDM are compared. As can be seen from Table 5, when EN and GE are effectively hybridized, the fire risk of GN is small, which is equivalent to a low fire risk. This is because GN combustion produces a large amount of residual char with a high degree of graphitization, thus forming a dense carbon layer. Moreover, the residual char produced by EN is fibrous and effectively cross-linked with the aromatic residue char contributed by graphene to strengthen the carbon layer.

Usually in a fire, most fire casualties are caused by asphyxiation caused by smoke, so it is very important to reduce smoke generation and emission during combustion [25]. The dynamic smoke generation behavior of EPDM composites is characterized by total smoke generation (TSP), the total smoke rate (TSR) and the smoke generation rate (RSR), as shown in Figure 4c–e. The data are summarized in Table 5. As shown in Figure 4c, compared with EPDM, the TSP, TSR and RSR values of EPDM/AEG composites are significantly reduced. The TSP and TSR values of EPDM/AEG40 are especially reduced from 15.2 m^2^ for neat EPDM to 9.9 m^2^ for TSP, with a decrement of 34.9%, and reduced from 1715.2 m^2^/m^2^ for neat EPDM to 1124.5 m^2^/m^2^ for TSR, with a decrement of 34.4%, indicating that AEG is an effective flame retardant and smoke suppressor. Evidently, EN has an obvious effect on reducing the smoke release rate, resulting in a reduced TSP and TSR. The P-SPR of EPDM/AEG40 composite is also 69.4% lower than that of neat EPDM. It can be seen that the AEG flame-retardant system has an obvious smoke suppression effect by acting as a physical barrier to limit the transfer of smoke and dust. The slow release of heat and smoke is very beneficial for fire control and the escape of people caught in the fire. In addition, compared with neat EPDM, the reductions in the carbon monoxide and carbon dioxide generation rates of EPDM/AEG composites (Figure 5) also directly prove the higher char-forming ability of AEG. Notably, the peak release of CO for the EPDM/AEG40 is just 0.0057 g/s. This is a considerable flame-retardant index because CO can cause suffocation in a fire, which is the most common cause of death in a fire. As a result of high emissions of carbon dioxide, global warming is presenting a range of harms to humans, and AEG shows excellent performance in inhibiting CO_2_ release. The measured average effective hot comb (EHC) further provides a better understanding of the reduced flammability caused by coke formation [26]. The thermal combustion performance of EPDM/AEG composites (Table 5) proves the synergistic improvement in thermal stability and flame retardancy. In this regard, we conclude that GN acts as a carbon source in the AEG system and plays a very good role in preventing heat and flammable volatiles in the condensed phase.

### 3.4. Morphology of Intumescent Char Layer

In order to clarify the influence of char residue on the combustion of EPDM composite materials, the morphology of the residual chars after combustion are studied by SEM. Figure 6 shows the SEM images of the char residues of neat EPDM, EPDM/AG, EPDM/AEG and EPDM/AE composites after the cone calorimeter test. For the char residue of neat EPDM, a loose char layer can be clearly observed, along with no expansion, slight melting and shedding (Figure 6a). This is due to the lack of a char-forming agent, the insufficient thickness and the unbreakable strength of the carbon residue. The char residue of EPDM/AEG composites (Figure 6b,c,f) is more compact and complete, and with the increase in the amount of APP, the increase in acid source makes the char layer expand more obviously. Furthermore, to learn more about flame-retardant effect of EN and GE combined use, the char residues of EPDM/AG40, EPDM/AE40 and EPDM/AEG40 are studied by SEM images (Figure 6d–f). For the residues of EPDM/AG40 and EPDM/AE40, there is a large number of cracks and visible holes, which have a unavoidable negative effect on the barrier action of these residues; however, the residue of EPDM/AEG40 is more continuous, and there is almost no visible holes. Under high magnification conditions (Figure 6(d-2,e-2,f-2)), the degree of contrast is more obvious. For EPDM/AEG40, not only are the flammable and oxygen gasses blocked from the flame by its superior char layer, but the heat transfer process is prevented, resulting in a better flame retardancy for the EPDM/AEG40. 

SEM-EDS analysis was used to study the distribution and content of flame-retardant elements in the carbon residue. In Figure 7, the element mapping of the EPDM/AEG composites (Figure 7b,d,f) reveals changes in the content of carbon, nitrogen, oxygen and phosphorus atoms on the carbon surface. As shown in Table 6, compared with neat EPDM, EPDM/AEG composites have an increased carbon atom content and a decreased oxygen atom content, which further illustrates the ability of AEG to promote residue formation during combustion. The increase in phosphorus atoms further confirms the high phosphorus residue after combustion, which provides higher resistance during pyrolysis and combustion.

### 3.5. Static and Dynamic Mechanical Properties of EPDM/AEG Composites

The physical properties are a non-ignorable performance index for elastic rubber in almost all applications [27]. It should be emphasized that, unfortunately, there is a tendency for most flame retardants to remarkably damage the mechanical behavior of rubber, especially for nonpolar rubber. In most cases, polarity differences between polar fire retardants and nonpolar rubber result in the poor particle dispersion and compatibility that are responsible for these negative effects. However, an abnormal phenomenon occurs in the AEG system in that AEG has a positive effect on the rubber's mechanical properties, which is an urgent problem of the commonly used flame retardants needing to be solved. As shown in Figure 8, tensile strength and tear strength of EPDM/AEG composites are much higher than neat EPDM and EPDM/APP40, even for EPDM/AEG40 with a V-0 flame retardant rating, which has a tensile strength up to 33.0% and a tear strength up to 30.1% based on neat EPDM. However, while the tensile strength and tear strength of the EPDM/AEG composites both show a tendency to decrease increase in the APP content, they are always higher than the neat EPDM. For, EPDM/APP40 with a V-0 flame retardant rating, the tensile strength goes down to 29.8% compared to neat EPDM (Appendix A). Obviously, GN can compensate for the loss of mechanical properties caused by APP. Under the load of 40 phr APP, the tensile strength for EPDM/AE40 and EPDM/AG40 is also improved, resulting in EPDM/AE40 and EPDM/AG40 being 23.5% and 12.1% higher than neat EPDM, respectively. Moreover, what is more exciting is that AEG has a toughening effect as well as a strengthening effect on EPDM. By contrast, after effective hybridization between EN and GE, the tensile strength and tear strength of EPDM/AEG40 are both superior to EPDM/AE40 and EPDM/AG40, which is attributed to the synergistic effect of AEG caused by the reinforcing effect of the two-dimensional graphene with a high specific surface area and the one-dimensional Entermorpha fiber with a high aspect ratio. In order to explore the dynamic mechanical properties of EPDM/AEG composites, dynamic mechanical analysis was used. Figure 9a shows that the addition of AGE has little effect on the glass transition temperature (T_g_) of vulcanized EPDM. It is well dispersed and has little effect on the movement and internal friction of the EPDM molecular chain. According to Figure 9b, the initial values of E’ of EPDM/AEG30 and EPDM/AEG40 are significantly higher than those of EPDM/AG40 and EPDM/AE40, indicating that AEG has stronger reinforcing ability than AG and AE systems. It can also be seen that the storage modulus of EPDM increases with the increase in filler, which is attributed to the reinforcing effect of EN. The tan delta values at 0 °C of EPDM/AEG30 and EPDM/AEG40 are similar to EPDM/AE40 but significantly higher than that of EPDM/AG40, which means EPDM/AEG30, EPDM/AEG40 and EPDM/AE40 have better wet-skid resistance. The results indicate that the contribution of Enteromorpha in flame-retardant systems is critical to the wet-skid resistance of NR composites [28].

### 3.6. Flame Retardancy Mechanism

Based on the above analysis, the flame retardancy mechanism of the AEG system in the EPDM matrix is assumed in Figure 10. First of all, for the condensed phase, in the early degradation process, the premature decomposition of EN absorbs part of the heat, which causes the ignition time of the polymer to be delayed [28]. GE sheets with excellent thermal stability act as quality barrier to inhibit the penetration of combustible gas [29,30]. At the same time, the catalytic carbonization of metaphosphoric acid (HPO_4_) decomposed by the APP adsorbed on the graphene surface leads to the production of a great amount of phosphorous and carbonaceous carbon residues [31,32], which is also confirmed from the EDS data shown in Figure 7 and Table 5. In addition, since EN maintains a good fiber morphology after burning, the high aspect ratio of EN can hold the carbon particles together, and the ammonia gas released from the APP expands to form a honeycomb carbon layer to produce high strength on the internal material. Because of its relatively large specific surface area, TGE can play a role in reinforcing the carbon layer and capturing free radicals in the gas phase. Secondly, with regard to the gas phase, the ammonia decomposed from APP effectively dilutes the volatile during combustion. It can be seen that the tripartite cooperation mechanism of the AEG system is the main source of the excellent flame-retardant properties of EPDM composites. In order to confirm this hypothesis, the microstructure of coke slag was analyzed by Raman spectroscopy.

Raman spectroscopy reveals the flame retardancy mechanism of AEG. Figure 11 and Appendix A show that peak fitting is performed for each spectrum to resolve the curve into D and G bands (Figure 11). The combined intensity (I_D_/I_G_) reflects the graphitization degree of residual carbon, and the lower the I_D_/I_G_ value is, the better the residual carbon structure is [33,34,35]. Obviously, the I_D_/I_G_ ratio in this study follows the sequence of EPDM/AEG40 < EPDM/AE40 < EPDM/AG40, which indicates that EN can promote the formation of highly graphitized and insulating carbon layers, which is the main mechanism for suppressing flammability.

## 4. Conclusions

The bio-based AEG flame retardants has a synergistic effect on smoke suppression and flame retardancy in EPDM due to the intumescent flame retardancy mechanism. The microstructure obtained from Raman and SEM-EDS analyses of the residual chars show that the AEG can significantly promote high graphitization and contains a phosphorus structure. Thus, flammability parameters and fire risk are effectively reduced. In addition, the AEG is beneficial for the mechanical properties of EPDM within a proper addition level. Even under high load conditions, EN and GE can compensate for the negative effects caused by the addition of flame retardants on mechanical properties.

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
