# Peer review of "The Study of Enteromorpha-Based Reinforcing-Type Flame Retardant on Flame Retardancy and Smoke Suppression of EPDM"

_polymers, 2022, doi:10.3390/polym15010055_

Round 1

Reviewer 1 Report

Please specify all the abbreviations. There is no information about what is EN and GN, which is crucial for this paper I guess. Without it, the proper understanding of the paper is impossible. 

Tale 1 should include applied formulations, variable does not provide any information.

Ultrafine treatment should be described proprely in  Experimental and materials section.

Please provide standard deviation values for particle size in Figure 1.

"T2max corresponds to the pyrolysis of oxygen-containing functional groups in EN" - please describe in detail, since different groups have different stability.

For better understanding please provide results of TGA for APP.

Please discuss the formulation impact on vulcanization performance more deeply than just stating that "Enteromorpha treated by conventional drying process has no effect on the vulcanization process of EPDM". It can be see that the vulcanization parameters are differing noticeably (over 10%) between samples. Therefore, presented results should be discussed.

Why formulations are presented in details in Table 4 and not Table 1? Then what is the role of Table 1? It does not make any sense.

What are the standard deviations of LOI values presented in Table 4?

Please define NC and describe the V-0, V-1 and V-2 parameters.

Please provide scale bars on SEM images.

Figure 7 indicates that Neat EPDM has more P than other samples. Then why Table 6 provides different results?

Figure 9 is hardly visible, please enhance it. 

Also, DMA results should be discussed more deeply.

Why such a difference in tan delta peak is noted between EPDM/AG40 and other samples ?

The discussion should include more references to the other literature works.

Author Response

Reviewer 1:

  1. Please specify all the abbreviations. There is no information about what is EN and GN, which is crucial for this paper I guess. Without it, the proper understanding of the paper is impossible.

Answer: Thanks very much for your comment. The information for EN, GN, AE and AG is supplied in page 3 and Page 6”.

  1. Tale 1 should include applied formulations, variable does not provide any information.

Answer: Thanks very much for your comment. Table 1 is changed to applied formulations.

Table 1 Batch compositions.

Samples

EPDM

CB

ZnO

SA

S

M

TT

AEG

AG

AE

Neat EPDM

100

40

5

1

1.5

0.5

1.5

-

-

-

EPDM/AEG20

100

40

5

1

1.5

0.5

1.5

20

-

-

EPDM/AEG30

100

40

5

1

1.5

0.5

1.5

30

-

-

EPDM/AG40

100

40

5

1

1.5

0.5

1.5

-

40

-

EPDM/AE40

100

40

5

1

1.5

0.5

1.5

-

-

40

EPDM/AEG40

100

40

5

1

1.5

0.5

1.5

40

-

-

  1. Ultrafine treatment should be described proprely in Experimental and materials section.

 Answer: Thanks very much for your comment. Ultrafine treatment is described in Experimental and materials 2.3 section.

  1. Please provide standard deviation values for particle size in Figure 1.

 Answer: Thanks very much for your comment. Standard deviation values for particle size is provided in Figure 1.

  1. "T2max corresponds to the pyrolysis of oxygen-containing functional groups in EN" - please describe in detail, since different groups have different stability.

Answer: Thanks very much for your comment. The chemical composition of EN is complex and varies with the change of sea area, mainly are carboxy group, hydroxyl group, carbonyls and ester groups.

  1. For better understanding please provide results of TGA for APP.

Answer: Thanks very much for your comment. the results of TGA for APP is provided in Fig. S1.

Fig. S1 TG curve (a) and DTG curve (b) of APP under N2 atmosphere.

  1. Please discuss the formulation impact on vulcanization performance more deeply than just stating that "Enteromorpha treated by conventional drying process has no effect on the vulcanization process of EPDM". It can be see that the vulcanization parameters are differing noticeably (over 10%) between samples. Therefore, presented results should be discussed.

Answer: Thanks very much for your comment. The formulation impact on vulcanization performance is discussed deeply, “The presence of water is fatal to vulcanization of rubber, however, the vulcanization performance EPDM composites shown in Table 3 present that Enteromorpha treated by conventional drying process has no effect on the vulcanization and scorching time of EPDM which the important parameter tc10 and tc90 in vulcanization property has not changed basically.”

  1. Why formulations are presented in details in Table 4 and not Table 1? Then what is the role of Table 1? It does not make any sense.

Answer: Thanks very much for your comment. Formulations are presented in Table 1.

  1. What are the standard deviations of LOI values presented in Table 4?

Answer: Thanks very much for your comment. The standard deviations of LOI values is presented in Table 4.

  1. Please define NC and describe the V-0, V-1 and V-2 parameters.

Answer: Thanks very much for your comment. NC, V-2, V-1 and V-0 rating represent flammable, flammable, refractory and non-flammable materials.

  1. Please provide scale bars on SEM images.

Answer: Thanks very much for your comment. Scale bars on SEM images are provided.

  1. Figure 7 indicates that Neat EPDM has more P than other samples. Then why Table 6 provides different results?

Answer: Thanks very much for your comment. Peak height in Figure 7 is not indicative of element content.

  1. Figure 9 is hardly visible, please enhance it.

Answer: Thanks very much for your comment. Figure 9 is changed.

  1. Also, DMA results should be discussed more deeply.

Answer: Thanks very much for your comment. DMA results are discussed more deeply on page 22 and 23. “Fig. 9 (a) shows that the addition of AGE has little effect on the glass transition temperature (Tg) of vulcanized EPDM. It is well dispersed and has little effect on the movement and internal friction of the EPDM molecular chain. According to Fig. 9 (b), the initial value of E’ of EPDM/AEG30 and EPDM/AEG40 is significantly higher than those of EPDM/AG40 and EPDM/AE40, indicating that AEG has stronger reinforcing ability than AG and AE systems. It can also be seen that the storage modulus of EPDM increases with the increase of filler, which is attributed to the reinforcing effect of EN. Tan delta at 0 ℃ of EPDM/AEG30 and EPDM/AEG40 has little difference with EPDM/AE40, that is significantly higher than that of EPDM/AG40, which means the better wet-skid resistance for EPDM/AEG30, EPDM/AEG40 and EPDM/AE40. The results indicate the contribution of graphene in flame-retardant systems is critical to wet-skid resistance of NR composites.”

  1. The discussion should include more references to the other literature works.

Answer: Thanks very much for your comment.  The discussion is revised to include more references to the other literature works.

Reviewer 2 Report

Dear Authors,

the use of "waste components" and the use of natural ressources have a double interest. So the proposed study is very interesting from this point of view.

In the introduction the context is well given but it would be intersting to give some information about the researches performed on the development of biobased intumescent system. Why the use of Enteromorpha is more interesting than other natural ressources? (In terms of chemical composition, performances and not only due to its availability in the coast of China).

One of the main improvement needed in this article is the part dedicated to experimental and material. The composition of the different samples is not clear. The supplementary files is confusing (In Table S1, only APP is in gram! and the other components ?). For each experiments please add the number of tests performed.

Please specify in the text what is EN? I supposed that is Enteromorpha but I checked the files and it is not specify.

The scheme 1 should also be improved and the strategy should be given and justified with litterature (why using graphene?)

1: EPDM,

2:EPDM + classical intumescent system (blowing agent+acid source = APP, and carbon source = enteromorpha)

3:EPDM + synergy (classical intumescent + graphene)

The title of part 3.1 is "The particle size distribution of EN" but there are also results of zeta potential measurements. The title should be revised. I am a little bit confused becasue you speaks about particle size distribution but there is no histogram showing the frequency of size particles. To explain the difference of zeta potential you indicate that it is probably due to the size of the particle (Improvement of contact surface?). But if I well understand the EN have been modified with tanic acid and have hydroxyl group. If it is the case, has the number of OH-functions been controlled? Maybe the zeta potential measurements have been performed on "virgin enteromorpha"? That why it is very important to clarify the term of "EN".

In the part 3.2 you should first introduce the number of decomposition steps for each component (EN, AEG40). The TGA of APP and other based components should be added. If it is possible it would be interesting to have TGA-FTIR measurements to have an idea of chemaicla composition of released gases. In line 163-165 it is mentionned that the components have a good thermal stability (consistent with rubber processing). But TGA experiments have been performed under inert atmosphere. Is it the same atmosphere used for rubber processing?

The table3 and the performances of EDPM composites after vulcanization shlud be more explained (tc10/s ?, etc.)

In table 4, more details should be added for UL-94. Please add T1(s), T2(s), Total time combustion and not only the rating (NV, V-1, etc.)

Line 338-341 an hypothesis of synergistic effectic linked with surface area  is proposed to explain the mechanical properties. I think it will be very interesting to have SEM images to analyse the distribution of particles in EPDM.

I appreciate that you propose a flame retarded mechanism.

Concerning the form of this article, even if I am not a native, I suggest to revise the article because some sentences are confusing.

Author Response

The use of "waste components" and the use of natural ressources have a double interest. So the proposed study is very interesting from this point of view.

In the introduction the context is well given but it would be intersting to give some information about the researches performed on the development of biobased intumescent system. Why the use of Enteromorpha is more interesting than other natural ressources? (In terms of chemical composition, performances and not only due to its availability in the coast of China).

  1. One of the main improvement needed in this article is the part dedicated to experimental and material. The composition of the different samples is not clear. The supplementary files is confusing (In Table S1, only APP is in gram! and the other components ?). For each experiments please add the number of tests performed.

Answer: Thanks very much for your comment. The composition of the different samples is clearly shown in Table 1. In Table S1, all components are in gram.

Table S1 The batch compositions for AEG system.

Fire retardant

APP(g)

EN(g)

GE(g)

TA(g)

AEG20

20

20

1

0.1

AEG30

30

20

1

0.1

AG40

40

--

1

0.1

AE40

AEG40

40

40

20

20

--

1

--

0.1

  1. Please specify in the text what is EN? I supposed that is Enteromorpha but I checked the files and it is not specify.

Answer: Thanks very much for your comment. The information for EN, GN, AE and AG is supplied in page 3 and Page 6”.

  1. The scheme 1 should also be improved and the strategy should be given and justified with litterature (why using graphene?)

Answer: Thanks very much for your comment. What graphene does is it creates a barrier layer inside the composite to block flammable and toxic gases, and graphene also acts as a reinforcing agent for the residue char layer.

1: EPDM,

2:EPDM + classical intumescent system (blowing agent+acid source = APP, and carbon source = enteromorpha)

3:EPDM + synergy (classical intumescent + graphene)

The title of part 3.1 is "The particle size distribution of EN" but there are also results of zeta potential measurements. The title should be revised. I am a little bit confused becasue you speaks about particle size distribution but there is no histogram showing the frequency of size particles. To explain the difference of zeta potential you indicate that it is probably due to the size of the particle (Improvement of contact surface?). But if I well understand the EN have been modified with tanic acid and have hydroxyl group. If it is the case, has the number of OH-functions been controlled?

  1. Maybe the zeta potential measurements have been performed on "virgin enteromorpha"? That why it is very important to clarify the term of "EN".

Answer: Thanks very much for your comment. The particle size of EN shown in Fig.1a is D50. The zeta potential measurements in Fig.1B has been performed on "virgin enteromorpha" rather than TA modified enteromorpha. EN is the abbreviation of Enteromorpha.

  1. In the part 3.2 you should first introduce the number of decomposition steps for each component (EN, AEG40). The TGA of APP and other based components should be added. If it is possible it would be interesting to have TGA-FTIR measurements to have an idea of chemaicla composition of released gases. In line 163-165 it is mentionned that the components have a good thermal stability (consistent with rubber processing). But TGA experiments have been performed under inert atmosphere. Is it the same atmosphere used for rubber processing?

Answer: Thanks very much for your comment. The TGA of APP is added in Fig.S1.

Fig. S1 TG curve (a) and DTG curve (b) of APP under N2 atmosphere.

  1. The table3 and the performances of EDPM composites after vulcanization shlud be more explained (tc10/s ?, etc.)

Answer: Thanks very much for your comment.The performances of EDPM composites after vulcanization including tc10 and tc90 are more explained on page 10.

  1. In table 4, more details should be added for UL-94. Please add T1(s), T2(s), Total time combustion and not only the rating (NV, V-1, etc.)

Answer: Thanks very much for your comment. T1(s), T2(s) are added for UL-94 shown in Table 4, and not only the rating (NV, V-1, etc.)

Table 4 LOI and UL-94 experimental data for EPDM rubber composites.

Samples

t1

t2

LOI

(%)

UL-94

Dripping

Neat EPDM

45

78

24.1+0.2

NC

Yes

EPDM/AEG20

39

69

28.8+0.4

NC

Yes

EPDM/AEG30

8

12

30.1+0.4

V-1

No

EPDM/AG40

15

20

29.5+0.3

V-2

Yes

EPDM/AE40

13

18

31.2+0.5

V-1

No

EPDM/AEG40

4

10

32.5+0.5

V-0

No

  1. appreciate that you propose a flame retarded mechanism.

Answer: Thanks very much for your comment.

  1. Concerning the form of this article, even if I am not a native, I suggest to revise the article because some sentences are confusing.

Answer: Thanks very much for your comment. The manuscript is revised especially in grammar and expression.

Reviewer 3 Report

Dear,

The authors developed composites with flame-retardant potential. The manuscript has merit for publication. I suggest minor revisions:

> The abstract needs to be revised. For example, the authors must present the main findings and results, the technological potential, and the experimental methodology;

> The introduction needs to clarify the novelty of the manuscript. In addition, the authors must mention the importance of the environment and the possibility of reintroduction in the production chain;

> Characterization. “The specimen dimension is 130*6.5*3.2 mm3; The 100 dimension of samples is 130*13*3.2 mm3; The dimension of samples is 100*100*3.2 mm3”. Please correct the multiplication sign;

> Please add the experimental error from Figure 8(b);

Author Response

The authors developed composites with flame-retardant potential. The manuscript has merit for publication. I suggest minor revisions:

  1. The abstract needs to be revised. For example, the authors must present the main findings and results, the technological potential, and the experimental methodology;

Answer: Thanks very much for your comment. The abstract is revised,”Enteromorpha as a waste from marine pollution brings great pressure to environmental governance every year, especially for China. Under the premise of shortage of industrial materials, taking appropriate measures can turn waste into wealth, which will benefit us a lot. In this work, a bio-based reinforcing type flame retardant based on Enteromorpha is designed. The designed Enteromorpha based flame retardant system mainly focused on the reinforcing action and the flame retardant effect on ethylene-propylene-diene tripolymer (EPDM) is researched systemically. The results show that When 40 phr AEG is added, the LOI of EPDM/AEG40 reaches 32.5% and the UL-94 reaches V-0 level. The PHRR and THR values of EPDM/AEG40 are 325.9 kW/m2 and 117.6 MJ/m2 with a decrement of 67.3% and 29.7% compared with the results of neat EPDM composite respectively. Especially, the TSP and TSR values of EPDM/AEG40 is reduced from 15.2 m2 of neat EPDM to 9.9 m2 with a decrement of 34.9 % and reduced from 1715.2 m2/m2 of neat EPDM to 1124.5 m2/m2 with a decrement of 34.4 %, indicating that AEG is effective in flame retardancy and smoke suppression. Meanwhile, tensile strength and tear strength of EPDM/AEG composites are much higher than neat EPDM, therefore, with the development of innovate reinforcing type flame-retardant Enteromorpha in the future, the application of Enteromorpha in the polymer flame-retardant field will surely usher in bright development.”

  1. The introduction needs to clarify the novelty of the manuscript. In addition, the authors must mention the importance of the environment and the possibility of reintroduction in the production chain;

Answer: Thanks very much for your comment. The introduction is modified to clarify the novelty of the manuscript. In addition, the importance of the environment and the possibility of reintroduction in the production chain is mentioned on page 4: “In this work, we design another efficient bio-based reinforcement type intumescent flame-retardant system (AEG) based on Enteromorpha which is environmental and possible for reintroduction in polymer flame retardant field. For the AEG system, ammonium polyphosphate (APP) serves as both an acid source and a gas source, the simple hybrid material (GN) by loading graphene (GE) and Enteromorpha (EN) using tannic acid (TA) as regulator, which serving as an acid source and a carbonizing agent. The AEG intumescent flame-retardant system mainly focuses on the reinforcing action and the flame-retardant effect on ethylene-propylene-diene tripolymer (EPDM). “

  1. “The specimen dimension is 130*6.5*3.2 mm3; The 100 dimension of samples is 130*13*3.2 mm3; The dimension of samples is 100*100*3.2 mm3”. Please correct the multiplication sign;

Answer: Thanks very much for your comment. It is corrected.

  1. Please add the experimental error from Figure 8(b);

Answer: Thanks very much for your comment. Figure 8(b) is added the experimental error.

Reviewer 4 Report

The research aims to enhance the potential use of an alga, Enteromorpha, mainly present in the seas of China and which normally constitutes a waste with a high environmental impact. The authors, showing the flame retardant and reinforcing properties of these algae in a rubbery matrix, highlight their potential as bio-resources of useful raw materials.

The topics covered are interesting and the techniques for characterizing the materials studied are undoubtedly relevant.

However, the description of the contents of the manuscript is carried out in an often incomprehensible language. Adding to this consideration the numerous typos in the text, it is clear that this contribution cannot provide a significant advancement of knowledge at present.

For example, some points of the text that must be revised from a linguistic or even grammatical point of view are the following.

Paragraph "Introduction"

lines 22-25: The sentence starting with "Especially for ..." is unclear. Please check.

lines 38-40: usually in research articles you do not write in the first person. Please revise.

lines 43-45: the sentence starting with "Fiber with a proper ..." is not well constructed and, therefore, difficult to understand.

lines 55-57: please check the sentence. There are probably errors related to a "cut and paste" operation.

line 61: the abbreviation AEG is introduced in the text. For clarity of explanation towards interested readers, it is necessary to make explicit the meaning of the abbreviations, at least the first time they are introduced in the text.

lines 62-65: It is suggested to rewrite this sentence in a more understandable way.

line 68: correct the word "analyzed" as "analyzed".

Paragraph 2.1 "Materials" - Line 77: in the text it says "was supplied by the sixth element": What does it mean?

Paragraph 2.2 "Characterization"

Line 88: the text says "from a set wavelength": What does it mean?

Line 95: please correct the word "stain" as "strain".

Line 98: It is suggested to simplify "standard oxygen index test ASTM D2863" in "standard ASTM D2863" to avoid repetitions.

Paragraph 2.4 “Preparation of composite materials”

Lines 117-119: the sentence is poorly constructed. Please revise.

line 127: it is written "ram dia pressure". What does it mean?

lines 139-140: incorrectly written "of and". Please correct.

line 144: there is an "s" after the word "dispersity". Please cancel.

line 150: it is written "interaction of the filler-matrix interface". It is suggested to correct as "interaction at the filler-matrix interface".

Paragraph 3.2 “Thermal stability of EN and AEG40”

Lines 169-172: Sentence long and poorly constructed. Please check and rephrase.

Lines 172-175: Some considerations.

Paragraph 3.3 “Flame retardant properties of EPDM rubber composites”

Lines 204-209: Given the length of the sentence, it is advisable to divide it in at least two parts, maybe putting a full stop after the word "performance".

Lines 209-210: Unclear sentence. Please check.

Lines 238-239: Unclear sentence, please rephrase.

Lines 239-241: Long and poorly constructed sentence. Please correct.

And so on.

Therefore, MINOR REVISIONS are suggested aimed at an accurate rereading of the manuscript with the recommendation to review the language above all, hopefully under the supervision of a native speaker colleague.

Author Response

The research aims to enhance the potential use of an alga, Enteromorpha, mainly present in the seas of China and which normally constitutes a waste with a high environmental impact. The authors, showing the flame retardant and reinforcing properties of these algae in a rubbery matrix, highlight their potential as bio-resources of useful raw materials.The topics covered are interesting and the techniques for characterizing the materials studied are undoubtedly relevant.However, the description of the contents of the manuscript is carried out in an often incomprehensible language. Adding to this consideration the numerous typos in the text, it is clear that this contribution cannot provide a significant advancement of knowledge at present. For example, some points of the text that must be revised from a linguistic or even grammatical point of view are the following.

Paragraph "Introduction"

  1. lines 22-25: The sentence starting with "Especially for ..." is unclear. Please check.

Answer: Thanks very much for your comment. The sentence is modified as following “Especially for the rubber industry, rubber as an important commercial material is inseparable from our lives. For engineering applications, adding filler to rubber is a necessary condition to improve its modulus, strength, wear resistance and fatigue resistance and other properties.”

  1. lines 38-40: usually in research articles you do not write in the first person. Please revise.

Answer: Thanks very much for your comment. The sentence is modified as following: “Qingdao is a beautiful city in China where every year a large amount of enteromorpha needs to be cleaned off the coast. In 2015, more than 70,000 tons of seaweed were treated.”

  1. lines 43-45: the sentence starting with "Fiber with a proper ..." is not well constructed and, therefore, difficult to understand.

Answer: Thanks very much for your comment. The sentence is modified as following: “In rubber industry, fiber with a certain aspect ratio is usually used to improve the dimensional stability of unvulcanized rubber and the comprehensive properties of vulcanizated rubber by increasing the green strength of rubber matrix.”

  1. lines 55-57: please check the sentence. There are probably errors related to a "cut and paste" operation.

Answer: Thanks very much for your comment. The sentence is modified on page 4.

  1. line 61: the abbreviation AEG is introduced in the text. For clarity of explanation towards interested readers, it is necessary to make explicit the meaning of the abbreviations, at least the first time they are introduced in the text.

Answer: Thanks very much for your comment. The meaning of the abbreviations for AEG, AE, AG is making in abstract section.

  1. lines 62-65: It is suggested to rewrite this sentence in a more understandable way.

 Answer: Thanks very much for your comment. It is corrected, as follows: “In the AEG system, ammonium polyphosphate (APP) is used as acid source and air source, the hybrid material (GN) prepared by GE and EN using TA as regulator is acid source and carbon source. AEG flame retardant system is mainly focused on the enhancement and flame retardant effects of ethylene-propylene-diene rubber (EPDM).”

  1. line 68: correct the word "analyzed" as "analyzed".

Answer: Thanks very much for your comment.  It is corrected.

  1. Paragraph 2.1 "Materials" - Line 77: in the text it says "was supplied by the sixth element": What does it mean?

Answer: Thanks very much for your comment. It is the name of company which supplying graphene to us.

Paragraph 2.2 "Characterization"

  1. Line 88: the text says "from a set wavelength": What does it mean?

Answer: Thanks very much for your comment. It is modified to “Raman spectra analysis were recorded using a Bruker FRS-100S with high-resolution  and a CCD detector in the wavelength range from 600 to 2500 cm-1. ”

  1. Line 95: please correct the word "stain" as "strain".

Answer: Thanks very much for your comment. It is corrected.

  1. Line 98: It is suggested to simplify "standard oxygen index test ASTM D2863" in "standard ASTM D2863" to avoid repetitions.

 Answer: Thanks very much for your comment.  It is corrected.

Paragraph 2.4 “Preparation of composite materials”

  1. Lines 117-119: the sentence is poorly constructed. Please revise.

Answer: Thanks very much for your comment. The sentence is revised, as follow: “Flame retardant AEG was prepared by compounding the prepared GN with APP in a certain proportion (Table S1), and EPDM composites were charged in a 200 ml Banbury mixer at 60 rpm rotor speed under 120 °C, the formula shown in Table 1.”

  1. line 127: it is written "ram dia pressure". What does it mean?

Answer: Thanks very much for your comment. The sentence is revised, as follow: “The sheets were pressed in a compression molding machine at 165°C for times equal to the optimal curing time and the pressure of 3.94·104 kg/m2.”

  1. lines 139-140: incorrectly written "of and". Please correct.

Answer: Thanks very much for your comment.  It is corrected.

  1. line 144: there is an "s" after the word "dispersity". Please cancel.

Answer: Thanks very much for your comment. It is corrected.

  1. line 150: it is written "interaction of the filler-matrix interface". It is suggested to correct as "interaction at the filler-matrix interface".

Answer: Thanks very much for your comment. It is corrected.

Paragraph 3.2 “Thermal stability of EN and AEG40”

  1. Lines 169-172: Sentence long and poorly constructed. Please check and rephrase.

Answer: Thanks very much for your comment. The sentence is revised, as follow: “Hongsheng Liu et al. [25] have studied that Enteromorpha as fillers is not suitable for conventional polymers especially for rubber composites, since it not only it is hydrophilic, but also contains some moisture. This is not easy to remove completely, which is due to growing in salt water. The presence of water is fatal to vulcanization of rubber, however, the vulcanization results of EPDM composites (Table 3) present that Enteromorpha has no effect on the vulcanization and scorching time of EPDM because the important parameter tc10 and tc90 has not changed basically.”  

  1. Lines 172-175: Some considerations.

Answer: Thanks very much for your comment. The sentence is revised, as follow: “The presence of water is fatal to vulcanization of rubber, and the vulcanization results of EPDM composites (Table 3) present that Enteromorpha has no effect on the vulcanization and scorching time of EPDM because the important parameter tc10 and tc90 has not changed basically. The results show that Enteromorpha is almost dried completely.”

Paragraph 3.3 “Flame retardant properties of EPDM rubber composites”

  1. Lines 204-209: Given the length of the sentence, it is advisable to divide it in at least two parts, maybe putting a full stop after the word "performance".

Answer: Thanks very much for your comment. The sentence is revised.

  1. Lines 209-210: Unclear sentence. Please check.

Answer: Thanks very much for your comment. The sentence is revised, as follow: “In EPDM/AEG composites, AEG flame-retardant systems improve the flame-retardant performance of EPDM mainly by reducing the amount and rate of combustion heat release.”

  1. Lines 238-239: Unclear sentence, please rephrase.

Answer: Thanks very much for your comment. The sentence is revised, as follow: “As can be seen from Table 5, when EN and GE are effectively hybridized, the fire risk of GN is small, which is equivalent to a low fire risk.”

  1. Lines 239-241: Long and poorly constructed sentence. Please correct.

Answer: Thanks very much for your comment. The sentence is revised, as follow: “It is because GN combustion produces a large amount of residue char with a high degree of graphitization, thus forming a dense carbon layer. Moreover, the residue char produced by EN is fibrous and effectively cross-linked with the aromatic residue char contributed by graphene to strengthen the carbon layer.”

Round 2

Reviewer 1 Report

OK

Reviewer 2 Report

Dear Authors,

thanks a lot to take into account most of my comments.